# The devil is not as black as he is painted? On the positive relationship between food industry conspiracy beliefs and conscious food choices

**Marta Marchlewska** [1]*, **Dagmara Szczepańska**[1,2], **Adam Karakula**[1], **Zuzanna Molenda**[1], **Marta Rogoza**[1], **Dominika Maison**[3]

1 Institute of Psychology, Polish Academy of Sciences, Warsaw, Poland, 2 The Maria Grzegorzewska University, Warsaw, Poland, 3 Faculty of Psychology, University of Warsaw, Warsaw, Poland

* mmarchlewska@psych.pan.pl

## Abstract

Previous research found that conspiracy beliefs were usually activated when individuals faced different types of psychological threats and that they led mainly to maladaptive individual and societal outcomes. In this research, we assumed that potential harmfulness of conspiracy beliefs may depend on the context, and we focused on the link between food industry conspiracy beliefs and conscious food choices. We hypothesized that food industry conspiracy beliefs may allow for a constructive attempt to protect oneself against real or imagined enemies (i.e., food industry companies) by conscious food choices (e.g., paying attention to how much the food products are processed). We tested this hypothesis among Polish participants (Study 1; $N = 608$; cross-sectional and Study 2; $N = 790$; experimental). Study 1 confirmed that context-specific conspiracy beliefs (but not general notions of conspiracy) are associated with adaptive consumer behaviors. Study 2 showed that inducing feelings of threat related to the possibility of purchasing food contaminated by a harmful bacteria (vs. control condition) increased food industry conspiracy beliefs, which were further positively linked to conscious food choices. We discuss the role of threat and conspiracy beliefs in adaptive consumer behaviors related to food choices.

## Introduction

Extant literature points toward the phenomenon of responsible, conscious, and reflexive consumption, highlighting the fact that modern consumers are paying at least some degree of attention to such issues as ethics, product composition and origin [1], or the environmental impact of purchased goods [2–7]. Although some researchers provide evidence for that many product purchase decisions are unconscious and automatic [8, 9], others show that consumers' choices are not motivated by brand awareness or image [10, 11] but rather by health, environmental, or social reasons [12–14]. The latter also applies to food products and has even turned into a form of social pressure, put by the consumers on the food industry, to include relevant

**Funding:** This research was funded by National Science Centre Poland under Opus grant (2019/35/B/HS6/00123) awarded to MM. The funders had no role in study design, data collection and analysis, decision to publish, or preparation of the manuscript.

**Competing interests:** The authors have declared that no competing interests exist.

information on the packaging [15, 16]. In this paper, we analyze the psychological concomitants of this phenomenon, focusing on the role of food industry conspiracy beliefs in conscious food choices. Specifically, we examine whether a conviction that food industry companies are secretly conspiring against consumers may translate into psychological mobilization in the form of paying attention to the quality of purchased food products.

## Conspiracy beliefs and (mal)adaptive behaviors

Conspiracy beliefs are mostly framed in terms of beliefs in the existence of a "vast, insidious, preternaturally effective international conspiratorial network designed to perpetrate acts of most fiendish character" [17] (p. 14). By explaining how powerful and evil out-groups covertly influence or cause major world events, conspiracy beliefs usually lead to negative societal outcomes [18]. Previous research found, for example, that people who show a general tendency towards conspiracy theories are less willing to take part in conventional political activities (e.g., are less inclined to register to vote; [19, 20]). Different types of conspiracy beliefs were also positively correlated with anti-science attitudes [21], they were related to lower adherence to safety and self-isolation guidelines [22, 23], lower willingness to vaccinate against COVID-19 [24] or higher freeriding during the pandemic [23]. Conspiracy beliefs may also fuel extremism [25, 26] and lead to illegal actions, such as occupying buildings [27]. Moreover, previous research found conspiracy beliefs to predict prejudice, negative out-group attitudes, and violence [28, 29]. This is because adopting conspiratorial explanations is closely related to lower levels of trust, scapegoating, and projecting societal problems onto real or imagined enemies who can be blamed for individual or collective problems [30]. One may ask, however, whether conspiracy beliefs always must necessarily bring damaging consequences.

According to Krekó [31], there are situations when conspiracy beliefs could be useful and adaptive. For example, they may provide a sense of community for people with marginal views [32], open opportunities for political debate [33], or inspire people to mobilize toward collective goals with the intention to bring about social change [34]. It is worth noticing, however, that till now, mobilizing aspects of conspiracy beliefs were mainly explored in relation to group-level processes (e.g., collective action; [34]) with disregard to the individual perspective on this issue. At the same time, from an evolutionary perspective, higher suspicion, and sensitivity to clues of danger, associated with conspiracy beliefs, can be a sort of strategy that, while rising the frequency of false alarms, may decrease the probability of missing the threat by an individual (see signal detection model, [35]). As Robins and Post stated: "natural selection will favour animals that become sensitive to subtle clues of danger" [36] (p. 71).

In line with this logic, there are situations when conspiracy theories can be helpful in detecting different types of threat and further lead to mobilization and preparing strategies that can reduce the danger. In our work, we assume that this would refer to such conspiracy beliefs that draw our attention to potentially dangerous situations (e.g., poor food quality).

## Food industry conspiracy beliefs and conscious food choices

Nowadays food safety has become a concern for many societies [37], with specific cases of food and water contamination fuelling the perceived risk of the possibility to consume a harmful product [38]. According to data gathered by the Lloyd's Register Foundation [39], over 200 diseases (from diarrhoea to cancer) can be caused by unsafe food or water, containing harmful bacteria, viruses, parasites, chemical substances, or other contaminants. It is estimated that every year 600 million people become ill because of consuming unsafe products and 420 thousand die, especially in low and middle income countries [39]. Despite an increasing number of food safety regulations being introduced by local, as well as international organizations, such

as the British Food Safety Act (predecessor to the EU regulations) [40] or the Food Safety System Certification 22000 [41], instances of large-scale food scandals still occur. Arguably the largest scandal of this kind involved the Peanut Corporation of America and broke out in 2009, when 9 people died and over 10 thousand fell ill after consuming peanuts containing salmonella [42]. The case not only led to a massive recall of over 4000 different products in the US, but it also inspired a debate on state responsibility in facilitating unsafe conditions in food industry [42]. Another example is the European Union, where more than 90,000 cases of Salmonella are recorded each year and the main risk of infection in humans is associated with the consumption of contaminated food [43].

Although this data remains worrisome, not everyone declares behaviors encompassed by conscious consumption as typical for their regular food choices. For instance, Grunert and colleagues [44] found that even moderately high levels of concern about sustainability in food production did not translate into a specific motivation to use sustainability labels. On the other hand, previous research, largely inspired by the Protection Motivation Theory, showed that high levels of fear drove change in terms of both behavior and attitude towards health; for example, it inclined individuals to eat healthier food and physically exercise [45–48]. Given that one characteristic of conspiracy beliefs is exaggerating the direct threat specific choices may entail, by drawing attention to their potential dangers [35], a food industry conspiracy belief may also positively predict adaptive behaviors, at least on the declarative level. Therefore, in the present research, we aimed to explore the role of food industry conspiracy beliefs in shaping attitudes and behaviors related to conscious consumption in the area of food choices.

We define food industry conspiracy beliefs as convictions that agents responsible for the production, distribution, and safety inspection intentionally conceal certain facts regarding food products to fulfil their secret goals. Among the most commonly known food conspiracies are the belief that fluoride was deliberately added to drinking water during the Cold War to weaken the American people and make them "susceptible to a Communist takeover" [49] (p. 1559) or the theory that a United Nations sustainable development plan–Agenda 21 –intentionally uses genetically modified foods to make people fall ill and, by that, to decrease world population [50]. However, it needs to be highlighted that our intention was not to verify the validity of these accusations, but to explore the psychological concomitants of conspiracy beliefs related to the food industry. In line with our theorizing, these include higher susceptibility to external threats and higher motivation to protect oneself from the potential negative effects of these agents' actions by, for example, recurring to conscious food choices.

## Overview of the current research

The aim of our research was to investigate the prevalence of food industry conspiracy beliefs as well as factors associated with these beliefs. Previous research found that conspiracy beliefs are usually activated when facing different types of psychological threat [51] and lead mainly to maladaptive individual and societal outcomes [30, 52]. In this work we claim that concomitants of conspiracy beliefs may depend on the context so that in some cases belief in conspiracy theories may lead not only to negative consequences for the self [20, 30], but paradoxically, be associated with adaptive, healthy, behaviors. We claim that this is a "side effect" of some types of conspiracy theories, which not only warn people against real or imaginary enemies, but draw their attention to potentially dangerous, specific situations and, thus, decrease the probability of missing a threat [31, 35]. Importantly, the positive relationship between conspiracy beliefs and adaptive consumer choices should be present only in the case of context-specific (i.e., food industry) conspiracy beliefs and not generic conspiracist ideation (i.e., a belief

system which consists of a small number of generic, less specific, assumptions about the typicality of conspiratorial activity in the world; [53]).

In line with this logic, we assumed that food industry conspiracy beliefs (but not belief in general notions of conspiracy per se) should be associated with conscious food choices aimed at protecting one's own health (Study 1 and Study 2). Additionally, we assumed that inducing feelings of threat related to the possibility of purchasing food contaminated with a harmful bacteria should strengthen food industry conspiracy beliefs which, in turn, should be correlated with conscious food choices (Study 2). We tested these predictions in two studies conducted in Poland. Both studies included more than 400 participants, which gave us a power of .80 for detecting even small associations between variables (for $r$ = .14; [54]; G*Power yields a target of 395 participants).

Data for both studies was obtained via Pollster Institute–a Polish online research panel that has been previously used in academic studies (e.g., [22, 23]). Pollster has over 230,000 registered users. The studies were conducted on a non-probability, national quota sample of Poles representative for gender, age, settlement size and education. Quotas were based on the Central Statistical Office (GUS) data. Data was collected via Computer Assisted Web Interviews (CAWI). As a reward for taking part in the study, participants receive points that can be later monetized. Both studies were conducted in accordance with the Declaration of Helsinki and approved by the Research Ethics Committee of the Institute of Psychology, Polish Academy of Sciences (number of approval: 26/X/2020). Informed written consent was obtained from all subjects involved in the study. The data and code that support current findings and are necessary to replicate are openly available in Open Science Framework depository at https://osf.io/h4x5v/.

## Study 1

In Study 1 (cross-sectional), we sought to establish the basic relationship between food industry conspiracy beliefs and conscious food choices. To this end, we analysed data from a nationally representative study that included food industry conspiracy beliefs, conscious food choices, generic conspiracy beliefs, and demographics (age, gender, education, and settlement size). We assumed that food industry conspiracy beliefs (but not generic conspiracist ideation) should be positively related to conscious food choices.

### Method

**Participants and procedure.** Study 1 included a nationwide representative sample of Polish adults in terms of gender, age, completed level of education, and settlement size. The sample consisted of 603 respondents (329 women, 274 men), aged between 19 and 85 (M = 51.97, SD = 15.41). Data was collected on-line by a leading Polish online research panel that has been used in academic studies before [22, 23].

**Measures.** *Food industry conspiracy beliefs (short scale)*. The scale was developed for the purpose of the current study and was based on the characteristics of previous tools measuring conspiracist ideation [18, 19]. Each item included three elements: an implied agent (1) secretly undertaking specific action (2) to obtain some type of gain (3). To fit the context of the study, the agent was always associated with food industry companies and the actions were harmful for the consumers. It was measured with four items, asking about participants' beliefs about food industry conspiracies, using the following statements: "Food processing companies bribe quality controllers to hide the actual nutritional content of food products", "Food processing companies pay scientists to fabricate evidence for the innocuousness of ingredients that are in fact toxic", "Cases of food poisoning are being covered up so that food processing companies can keep on harming people with impunity", "Food processing companies secretly stuff foods

with harmful substances to earn more money". Participants responded on a scale from 1 = *I definitely disagree* to 5 = *I definitely agree*. The scale showed high reliability, α = .93. Exploratory Factor Analysis with principal axis extraction (Oblimin rotation) provided a single factor solution explaining 81.98% of the variance.

*Conscious food choices (short scale)*. The scale was developed for the purpose of the current study and was inspired by previous research on conscious consumption [4, 5, 7–9]. To adapt the scale to the conditions of the study, emphasis was placed on actions that can be undertaken while grocery shopping and that have been previously identified as conscious consumption. It was measured with three items, asking how much the respondents would be willing to do specific things during their next visit to a grocery shop, assessed by the following statements: "Before buying a food product, I will read the nutrition information displayed on the label", "Before buying a food product, I will pay attention to the country of origin of the groceries that I will be buying", "Before buying a food product, I will pay attention to how much the food products are processed". Participants were asked to determine their willingness to do these things on a scale from 1 = *I definitely will not do this* to 5 = *I will definitely do this*. The measure was internally consistent, α = .89. Exploratory Factor Analysis with principal axis extraction (Oblimin rotation) provided a single factor solution explaining 82.47% of the variance.

*General conspiracy beliefs*. Measured with the Generic Conspiracist Beliefs scale ([53]; Polish adaptation [55]). A total of 15 statements was applied, such as "Certain significant events have been the result of the activity of a small group who secretly manipulate world events", "New and advanced technology which would harm current industry is being suppressed". Participants responded on a scale from 1 = *definitely not true* to 5 = *definitely true*. The scale demonstrated very good reliability, α = .95.

*Covariates*. In addition to age and gender (coded Female = 0, Male = 1), participants were asked to indicate the highest level of education they had attained thus far (1 = *primary degree or no degree*, 2 = *vocational degree*, 3 = *high-school or post-secondary degree*, 4 = *university degree*) and settlement size (1 = *rural area*, 2 = *town up to 20 thousand residents*, 3 = *town between 20 and 99 thousand residents*, 4 = *town between 100 and 200 thousand residents*, 5 = *town between 200 and 500 thousand residents*, 6 = *city above 500 thousand residents*). Both education and settlement size were explanatory variables of categorical level. Thus, we decided to use a dummy coding procedure to control for their effects while predicting the variables of the main interest. Primary degree and rural area were used as reference categories.

**Statistical analyses.** Data was analyzed with IBM SPSS 27. Pearson product-moment correlation coefficient (Pearson's *r*) was used in correlation analyses. We also used hierarchical multiple linear regression analyses. Skewness and kurtosis analyses were conducted to assess the normality of the variables of interest. For food industry conspiracy beliefs skewness was -0.01 (*SE* = 0.10) and kurtosis was -0.44 (*SE* = 0.20), for conscious food choices skewness was -0.61 (*SE* = 0.10) and kurtosis was -0.08 (*SE* = 0.20), and for general conspiracy beliefs skewness was 0.10 (*SE* = 0.10) and kurtosis was -0.39 (*SE* = 0.20). There were no multicollinearity problems in our regression models, with all VIFs < 2.0 [56].

## Results and discussion

Since Study 1 used a nationally representative sample of Poles, we first explored the agreement with the food industry conspiracy items (Fig 1) by calculating the average percentage score for each answer to all items. Around 31.1% of all participants agreed with the statements arguing that the food industry is involved in some kind of conspiracy.

Next, we computed correlations between the variables. Conscious food choices were positively related to food industry conspiracy beliefs but unrelated to general conspiracy beliefs.

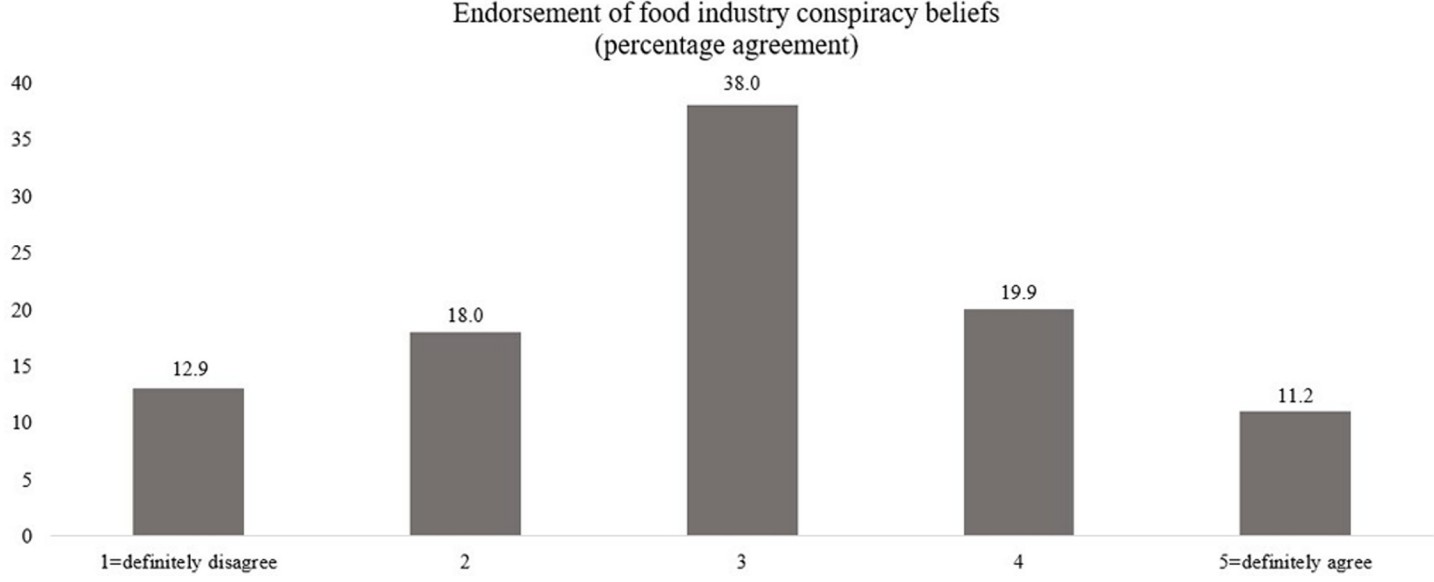

**Fig 1. Prevalence of food industry conspiracy beliefs in Poland.**

Both types of conspiracy beliefs (i.e., food industry conspiracy beliefs and general conspiracy beliefs) were positively correlated to each other. All means, standard deviations, and zero-order correlations can be found in Table 1. To investigate if there were any differences between genders (female = 0, male = 1), we computed an independent samples t-test for the main variables. There was no significant effect of gender (Food industry conspiracy beliefs, $t(601) = 0.70$, $p = .483$; Conscious food choices, $t(601) = 1.76$, $p = .079$; General conspiracy beliefs, $t(601) = 0.07$, $p = .946$).

We then performed a hierarchical multiple regression analysis to investigate whether food industry conspiracy beliefs (but not general conspiracy beliefs) would be positively related to conscious food choices when controlling for basic demographic variables (see Table 2). In Step 1, we introduced gender, age, education, and settlement size. We found a positive and significant effect of age on conscious food choices. In Step 2, we introduced food industry conspiracy beliefs and general conspiracy beliefs. Food industry conspiracy beliefs (but not general conspiracy beliefs) was a positive and significant predictor of conscious food choices. The positive effect of age remained significant.

Study 1 confirmed our prediction about the positive correlation between the endorsement of food industry conspiracy beliefs and conscious food choices. It also showed that context-

**Table 1. Means, standard deviations, and zero-order correlations (Study 1).**

| Measure | M | SD | 1 | 2 | 3 |
|---|---|---|---|---|---|
| 1. Food industry conspiracy beliefs | 2.99 | 1.05 | - | .11** | .62*** |
| 2. Conscious food choices | 3.66 | 1.04 | | - | .04 |
| 3. General conspiracy beliefs | 2.75 | 0.94 | | | - |

*Note*. We also conducted correlation analyses using Spearman test. Results remained the same.

*$p < .05$.

**$p < .01$.

***$p < .001$.

**Table 2. Predictors of conscious food choices (Study 1).**

| Variable | Step 1 | | | | Step 2 | | | |
|---|---|---|---|---|---|---|---|---|
| | $B$ | 95% CI | β | $p$ | $B$ | 95% CI | β | $p$ |
| | | LL　　UL | | | | LL　　UL | | |
| Gender (Female = 0, Male = 1) | -0.13 | [-0.30, 0.04] | -.06 | .130 | -0.12 | [-0.29, 0.04] | -.06 | .148 |
| Age | 0.01 | [0.01, 0.02] | .15 | < .001 | 0.01 | [0.004, 0.02] | .14 | < .001 |
| Vocational degree | -0.38 | [-0.95, 0.19] | -.13 | .187 | -0.40 | [-0.96, 0.17] | -.14 | .172 |
| High-school or post-secondary degree | -0.04 | [-0.58, 0.50] | -.02 | .886 | -0.05 | [-0.59, 0.49] | -.02 | .860 |
| University degree | 0.02 | [-0.52, 0.57] | .01 | .935 | 0.03 | [-0.51, 0.58] | .02 | .909 |
| Town up to 20 thousand residents | 0.08 | [-0.22, 0.38] | .02 | .597 | 0.09 | [-0.21, 0.38] | .03 | .564 |
| Town between 20 and 99 thousand residents | 0.09 | [-0.14, 0.31] | .04 | .449 | 0.09 | [-0.13, 0.32] | .04 | .409 |
| Town between 100 and 200 thousand residents | 0.10 | [-0.23, 0.42] | .03 | .567 | 0.10 | [-0.23, 0.43] | .03 | .550 |
| Town between 200 and 500 thousand residents | 0.02 | [-0.29, 0.32] | .004 | .925 | 0.01 | [-0.29, 0.32] | .004 | .926 |
| City above 500 thousand residents | 0.12 | [-0.15, 0.39] | .04 | .377 | 0.13 | [-0.14, 0.41] | .05 | .329 |
| General conspiracy beliefs | | | | | -0.02 | [-0.13, 0.10] | -.02 | .778 |
| Food industry conspiracy beliefs | | | | | 0.12 | [0.02, 0.22] | .12 | .018 |
| $R^2$ | | .05 | | | | .06 | | |
| $\Delta R^2$ | | | | | | .01* | | |
| $F$ | | $F_{(10, 592)} = 3.12$*** | | | | $F_{(12, 590)} = 3.29$*** | | |

*$p < .05$.
**$p < .01$.
***$p < .001$.

specific conspiracy beliefs (but not general notions of conspiracy) could be associated with adaptive consumer behaviors. These preliminary results suggested that in some cases conspiracy beliefs might be associated with adaptive behavior. People with a higher level of food industry conspiracy beliefs were found to be more conscious consumers and pay more attention to the composition of the food and its origin when shopping. Additionally, we found that higher age predicted more conscious food choices. It seems that older, more life-experienced individuals focus more on conscious choices while purchasing food products.

## Study 2

In Study 2, we aimed to replicate the pattern of results obtained in Study 1. Additionally, we aimed to examine the role of threat in boosting context-specific food industry conspiracy beliefs. Thus, we employed an experimental research design and tested a hypothesis that feelings of threat, related to the possibility of purchasing a food product that might contain a bacteria harmful to human health, would strengthen food industry conspiracy beliefs, which, in turn, would be associated with higher conscious food choices. We manipulated the feelings of threat with a short text about a case in which the Sanitary Inspectorate detected a harmful bacteria in many kinds of food products available for purchase in the most popular Polish supermarkets. As in Study 1, we controlled for basic demographics (age, gender, education, and settlement size) and general conspiracy beliefs to check whether the obtained effects are context-specific (i.e., related specifically to food issues).

One limitation of Study 1 was that we measured the crucial variables (e.g., food industry conspiracy beliefs and conscious food choices) with the use of short (four- and three-item) scales. Therefore, in Study 2 we examined whether the pattern of results obtained in Study 1 would conceptually replicate if we used better measurement tools. We operationalized the

food industry conspiracy beliefs and conscious food choices with 14 and 9 items respectively. The conceptual principles applied while developing the tools remained the same as in Study 1, with some items including an implicit allusion to the action secretly undertaken by food industry companies for their own benefit. Still, both extended versions of the scales showed high reliability (listed below) and Exploratory Factor Analysis provided single factor solutions for both of them. Additionally, in Study 2 we also controlled for individual differences related to consumer choices (i.e., frequency of grocery shopping and respondents' subjective financial situation).

## Method

**Participants and procedure.** As in Study 1, data for Study 2 was collected by an external research company, which has been used in academic studies before [22, 23], through Computer Assisted Web Interviews (CAWI). 790 respondents, aged between 18 and 83 ($M = 47.44$, $SD = 15.97$), participated in this study (419 women, 371 men). The sample was representative of Polish adults in terms of gender, age, completed level of education, and settlement size. Due to the experimental character of this study, we included an attention check–participants were asked about the content of the text that was used as a manipulation. Participants who failed the attention check ($n = 25$) were excluded from further analyses. This resulted in the final sample of 765 respondents (408 women, 357 men), aged between 18 and 83 ($M = 47.47$, $SD = 15.97$). When these participants were not excluded, the main pattern of results remained the same.

Participants were randomly assigned to one of two experimental conditions: threat ($n = 386$) and control ($n = 379$) by the research company. In both conditions, participants were exposed to a short, fabricated article, designed in such a way as to imitate an actual online article from a news portal. In the threat condition, it was an article about some food products that were withdrawn from stores due to the fact that sanitary authorities found a dangerous bacteria in them. In the control condition, they read an article providing advice on how to grow tomatoes at home. Afterwards, participants completed measures of conspiracy beliefs and conscious food choices. Then, they were asked to provide an answer to the attention check question. When the participants completed the questionnaire, they were debriefed and thanked.

**Measures.** *Food industry conspiracy beliefs (full scale)*. Measured with 14 items regarding food conspiracy beliefs: "Food processing companies secretly stuff foods with harmful substances to earn more money", "Nobody really knows what is inside of food products", "Food processing companies use genetically modified ingredients without letting the consumers know", "Cases of food poisoning are being covered up so that food processing companies can keep on harming people with impunity", "Harmful substances added to foods by food processing companies make them look good on the outside, despite being spoiled", "Food is stuffed with addictive substances to keep the customer loyal to it and to generate more profit", "Food processing companies bribe quality controllers to hide the actual nutritional content of food products", "Scientific evidence that some foods are harmful is being obscured by huge food processing companies", "Food processing companies pay scientists to fabricate evidence for the innocuousness of ingredients that are in fact toxic", "The real goal of food processing companies is high profit, regardless of the consequences for the consumers' health and life", "Food processing companies pay scientists to say that genetically modified food is healthy", "Artificially modified food allows food processing companies to control population size", "Food processing companies secretly add addictive substances to their products", "Food processing companies manipulate the amount of sugar in their products to make the consumers addicted to them". Participants responded on a scale from 1 = *definitely disagree* to 5 = *definitely agree*.

The measure demonstrated high reliability, α = .96. Exploratory Factor Analysis with principal axis extraction (Oblimin rotation) provided a single factor solution explaining 63.65% of the variance.

*Conscious food choices (full scale)*. Measured with nine items: "I will pay attention to the nutritional content of the food products that I will be buying", "Before buying a food product, I will read the nutrition information displayed on the label", "When choosing a food product, I will consider the nutrition information specified on the label", "Before buying a food product, I will pay attention to how much the food products are processed", "While shopping, I will use an app that will tell me which food products are healthy", "Before buying a food product, I will pay attention to the country of origin of the groceries that I will be buying", "I will buy groceries from local producers", "I will shop for groceries only in trusted places", "I will simply buy what I need, without analysing the nutritional content of the product (reverse coded item)". Participants were asked to determine whether they will do what the statement says using a scale from 1 = *I definitely will not do this* to 5 = *I will definitely do this*. The scale showed good reliability, α = .89. Exploratory Factor Analysis with principal axis extraction (Oblimin rotation) provided a single factor solution explaining 56.10% of the variance.

*General conspiracy beliefs*. As in Study 1, we used the Generic Conspiracist Beliefs scale ([53]; Polish adaptation [55]). There were 15 statements and participants responded on a scale from 1 = *definitely not true* to 5 = *definitely true*. The measure was internally consistent, α = .95.

*Covariates*. We used the same demographic variables as in Study 1: gender, age, level of education, and settlement size, but this time we also added a question about the respondents' subjective financial situation (1 = *definitely bad*, 2 = *bad*, 3 = *rather bad*, 4 = *average*, 5 = *rather good*, 6 = *good*, 7 = *definitely good*) and a question about the frequency of going grocery shopping (1 = *never*, 2 = *once a month*, 3 = *few times a month*, 4 = *once a week*, 5 = *two-three times a week*, 6 = *few times a week*, 7 = *everyday*). We followed the same procedure as in Study 1 to code dummy variables (i.e., level of education, and settlement size).

**Statistical analyses.** Data was analyzed with IBM SPSS 27. Mediation analyses were performed with Process v3.5 macro. Pearson product-moment correlation coefficient (Pearson's *r*) was used in correlation analyses. We used hierarchical multiple linear regression analyses. Similarly, as in Study 1, skewness and kurtosis analyses were conducted to assess the normality of the variables of interest in Study 2. For food industry conspiracy beliefs skewness was -0.14 (*SE* = 0.09) and kurtosis was -0.32 (*SE* = 0.18), for conscious food choices skewness was -0.50 (*SE* = 0.09) and kurtosis was -0.04 (*SE* = 0.18), and for general conspiracy beliefs skewness was 0.10 (*SE* = 0.09) and kurtosis was -0.65 (*SE* = 0.18). There were no multicollinearity problems in our regression models, with all VIFs < 2.0 [56].

## Results and discussion

First, we computed correlations between the variables. Conscious food choices were positively related to food industry conspiracy beliefs. We also found a significant, albeit weaker, correlation between conscious food choices and general conspiracy beliefs. Importantly, both types of conspiracy beliefs (i.e., food industry conspiracy beliefs and general conspiracy beliefs) were positively related to each other.

We also found that shopping frequency, and subjective financial situation were significantly positively related to conscious food choices. Food industry conspiracy beliefs were significantly negatively related to subjective financial situation. We also found that general conspiracy beliefs were negatively related to subjective financial situation. All means, standard deviations and zero-order correlations can be found in Table 3. To investigate if there were any differences between genders (female = 0, male = 1), we computed an independent samples t-test for

**Table 3. Means, standard deviations, and zero-order correlations (Study 2).**

| Measure | *M* | *SD* | 1 | 2 | 3 | 4 | 5 |
|---|---|---|---|---|---|---|---|
| 1. Food industry conspiracy beliefs | 3.08 | 0.93 | - | .18*** | .62*** | .07 | -.13*** |
| 2. Conscious food choices | 3.58 | 0.81 | | - | .09* | .14*** | .10** |
| 3. General conspiracy beliefs | 2.72 | 0.96 | | | - | -.07 | -.14*** |
| 4. Shopping frequency | 5.23 | 1.17 | | | | - | .09* |
| 5. Subjective financial situation | 4.40 | 1.10 | | | | | - |

*Note*. We also conducted correlation analyses using Spearman test. Results remained the same.

*p < .05.

**p < .01.

***p < .001.

the main variables. The effect of gender on food industry conspiracy beliefs ($t(763) = 1.17$, $p = .242$) and general conspiracy beliefs ($t(763) = 1.46$, $p = .146$) was non-significant. In case of conscious food choices, women ($M = 3.66$, $SD = 0.82$) scored higher than men ($M = 3.49$, $SD = 0.81$), $t(763) = 2.84$, $p = .005$.

Next, we computed a hierarchical regression analysis to investigate the effects of the experimental manipulation (threat vs. control) on food industry conspiracy beliefs (see Table 4). Experimental manipulation was coded: 0 = control condition and 1 = threat condition. To control for the socio-demographic variables, in Step 1 we included not only gender, age, education, settlement size, but also variables about the frequency of grocery shopping and the subjective financial situation. Age positively and significantly predicted food industry conspiracy beliefs. Subjective financial situation and higher level of education (vs. primary degree) were also significantly, albeit negatively, related to food industry conspiracy beliefs. Finally, we a

**Table 4. Predictors of food industry conspiracy beliefs (Study 2).**

| Variable | | Step 1 | | | | | Step 2 | | | |
|---|---|---|---|---|---|---|---|---|---|---|
| | *B* | 95% CI | | β | *p* | *B* | 95% CI | | β | *p* |
| | | LL | UL | | | | LL | UL | | |
| Gender (Female = 0, Male = 1) | -0.11 | [-0.24, 0.02] | | -.06 | .097 | -0.12 | [-0.25, 0.01] | | -.07 | .066 |
| Age | 0.01 | [0.001, 0.01] | | .10 | .009 | 0.01 | [0.001, 0.01] | | .08 | .031 |
| Vocational degree | -0.28 | [-0.74, 0.19] | | -.08 | .249 | -0.29 | [-0.76, 0.17] | | -.09 | .215 |
| High-school or post-secondary degree | -0.47 | [-0.89, -0.05] | | -.25 | .027 | -0.49 | [-0.91, -0.08] | | -.26 | .020 |
| University degree | -0.63 | [-1.05, -0.21] | | -.34 | .004 | -0.65 | [-1.07, -0.23] | | -.35 | .002 |
| Town up to 20 thousand residents | 0.33 | [0.09, 0.56] | | .10 | .007 | 0.36 | [0.12, 0.59] | | .11 | .003 |
| Town between 20 and 99 thousand residents | 0.10 | [-0.08, 0.28] | | .05 | .275 | 0.12 | [-0.06, 0.30] | | .06 | .182 |
| Town between 100 and 200 thousand residents | 0.06 | [-0.19, 0.31] | | .02 | .649 | 0.07 | [-0.18, 0.32] | | .02 | .564 |
| Town between 200 and 500 thousand residents | 0.09 | [-0.16, 0.33] | | .03 | .500 | 0.10 | [-0.15, 0.34] | | .03 | .434 |
| City above 500 thousand residents | -0.17 | [-0.38, 0.05] | | -.06 | .131 | -0.14 | [-0.35, 0.07] | | -.05 | .199 |
| Shopping frequency | 0.05 | [-0.01, 0.11] | | .06 | .078 | 0.05 | [-0.003, 0.11] | | .07 | .062 |
| Subjective financial situation | -0.09 | [-0.15, -0.03] | | -.11 | .003 | -0.09 | [-0.15, -0.04] | | -.11 | .002 |
| Condition (control = 0; threat = 1) | | | | | | 0.26 | [0.13, 0.39] | | .14 | < .001 |
| $R^2$ | | .08 | | | | | .09 | | | |
| $\Delta R^2$ | | | | | | | .01* | | | |
| *F* | | $F(12, 752) = 5.09^*$ | | | | | $F(13, 751) = 6.01^*$ | | | |

*p < .001.

found positive effect of living in town up to 20 thousand residents (vs. rural area) on conspiracy beliefs. In Step 2, we introduced variable coding experimental condition (threat vs. control). We found that experimental condition positively and significantly predicted food industry conspiracy beliefs: participants in the threat (vs. control) condition scored significantly higher on food industry conspiracy beliefs. The effects of age, place of residence, education and financial situation remained significant.

Then we computed a hierarchical regression analysis to investigate the effects of the experimental manipulation (threat vs. control) on general conspiracy beliefs (see Table 5). Experimental manipulation was coded 0 = control condition and 1 = threat condition. Again, in Step 1 we introduced the socio-demographic variables: gender, age, education, settlement size, as well as variables about the frequency of grocery shopping and the subjective financial situation. We found that age and subjective financial situation were significantly negatively related to general conspiracy beliefs. In turn, shopping frequency and living in smaller towns (vs. rural area) positively and significantly predicted the dependent variable. In Step 2, we introduced variable coding experimental condition (threat vs. control). In line with our assumptions, we did not find a significant effect of the experimental condition on general conspiracy beliefs. Effects of age, shopping frequency, subjective financial situation, and living in smaller towns remained significant.

Finally, we computed a hierarchical regression analysis to investigate the effects of the experimental condition, food industry conspiracy beliefs, and general conspiracy beliefs on conscious food choices (Table 6). The experimental manipulation was coded 0 = control condition and 1 = threat condition. In Step 1, we introduced the socio-demographic variables: gender, age, education, settlement size, as well as variables about the frequency of grocery shopping and the subjective financial situation. Gender was significant and negative predictor of conscious food choices. Age, shopping frequency, and subjective financial situation were

**Table 5. Predictors of general conspiracy beliefs (Study 2).**

| Variable | B | Step 1 95% CI | β | p | B | Step 2 95% CI | β | p |
|---|---|---|---|---|---|---|---|---|
| | | LL     UL | | | | LL     UL | | |
| Gender (Female = 0, Male = 1) | -0.09 | [-0.23, 0.05] | -.05 | .190 | -0.10 | [-0.23, 0.04] | -.05 | .167 |
| Age | -0.01 | [-0.01, -0.003] | -.13 | .001 | -0.01 | [-0.01, -0.003] | -.13 | < .001 |
| Vocational degree | -0.03 | [-0.51, 0.45] | -.01 | .911 | -0.04 | [-0.52, 0.45] | -.01 | .884 |
| High-school or post-secondary degree | -0.22 | [-0.65, 0.21] | -.12 | .312 | -0.23 | [-0.66, 0.20] | -.12 | .293 |
| University degree | -0.40 | [-0.84, 0.03] | -.21 | .069 | -0.41 | [-0.85, 0.02] | -.21 | .063 |
| Town up to 20 thousand residents | 0.31 | [0.07, 0.56] | .10 | .011 | 0.33 | [0.09, 0.57] | .10 | .008 |
| Town between 20 and 99 thousand residents | 0.19 | [0.01, 0.38] | .08 | .043 | 0.20 | [0.02, 0.39] | .09 | .034 |
| Town between 100 and 200 thousand residents | 0.04 | [-0.22, 0.30] | .01 | .738 | 0.05 | [-0.21, 0.31] | .02 | .700 |
| Town between 200 and 500 thousand residents | 0.05 | [-0.21, 0.30] | .01 | .724 | 0.05 | [-0.20, 0.31] | .02 | .691 |
| City above 500 thousand residents | -0.21 | [-0.43, 0.02] | -.07 | .067 | -0.19 | [-0.41, 0.03] | -.07 | .084 |
| Shopping frequency | 0.07 | [0.01, 0.12] | .08 | .024 | 0.07 | [0.01, 0.13] | .08 | .021 |
| Subjective financial situation | -0.13 | [-0.20, -0.07] | -.15 | < .001 | -0.14 | [-0.20, -0.07] | -.15 | < .001 |
| Condition (control = 0; threat = 1) | | | | | 0.12 | [-0.02, 0.25] | .06 | .085 |
| $R^2$ | | .08 | | | | .09 | | |
| $\Delta R^2$ | | | | | | .01 | | |
| F | | $F(12, 752) = 5.66^*$ | | | | $F(13, 751) = 5.47^*$ | | |

$^*p < .001.$

**Table 6. Predictors of conscious food choices (Study 2).**

| Variable | Step 1 | | | | Step 2 | | | | Step 3 | | | |
|---|---|---|---|---|---|---|---|---|---|---|---|---|
| | B | 95% CI | β | p | B | 95% CI | β | p | B | 95% CI | β | p |
| | | LL      UL | | | | LL      UL | | | | LL      UL | | |
| Gender (Female = 0, Male = 1) | -0.19 | [-0.30, -0.07] | -.12 | .001 | -0.19 | [-0.30, -0.08] | -.12 | .001 | -0.17 | [-0.28, -0.06] | -.11 | .003 |
| Age | 0.01 | [0.01, 0.02] | .26 | < .001 | 0.01 | [0.01, 0.02] | .26 | < .001 | 0.01 | [0.01, 0.02] | .25 | < .001 |
| Vocational degree | -0.18 | [-0.58, 0.23] | -.06 | .390 | -0.18 | [-0.58, 0.22] | -.06 | .381 | -0.14 | [-0.54, 0.26] | -.05 | .483 |
| High-school or post-secondary degree | -0.08 | [-0.45, 0.28] | -.05 | .651 | -0.09 | [-0.45, 0.28] | -.05 | .636 | -0.02 | [-0.38, 0.34] | -.01 | .918 |
| University degree | -0.13 | [-0.49, 0.24] | -.08 | .496 | -0.13 | [-0.49, 0.23] | -.08 | .483 | -0.04 | [-0.40, 0.33] | -.02 | .842 |
| Town up to 20 thousand residents | 0.04 | [-0.17, 0.24] | .01 | .728 | 0.04 | [-0.16, 0.25] | .02 | .685 | -0.01 | [-0.21, 0.19] | -.01 | .903 |
| Town between 20 and 99 thousand residents | -0.03 | [-0.19, 0.12] | -.02 | .673 | -0.03 | [-0.19, 0.13] | -.02 | .712 | -0.05 | [-0.21, 0.11] | -.03 | .520 |
| Town between 100 and 200 thousand residents | 0.06 | [-0.15, 0.28] | .02 | .570 | 0.07 | [-0.15, 0.28] | .02 | .553 | 0.06 | [-0.16, 0.27] | .02 | .615 |
| Town between 200 and 500 thousand residents | 0.03 | [-0.18, 0.24] | .01 | .780 | 0.03 | [-0.18, 0.25] | .01 | .762 | 0.02 | [-0.19, 0.23] | .01 | .859 |
| City above 500 thousand residents | 0.02 | [-0.17, 0.20] | .01 | .851 | 0.02 | [-0.16, 0.21] | .01 | .810 | 0.05 | [-0.14, 0.23] | .02 | .623 |
| Shopping frequency | 0.07 | [0.02, 0.11] | .10 | .007 | 0.07 | [0.02, 0.12] | .10 | .007 | 0.06 | [0.01, 0.11] | .08 | .017 |
| Subjective financial situation | 0.09 | [0.04, 0.14] | .13 | < .001 | 0.09 | [0.04, 0.14] | .12 | < .001 | 0.11 | [0.06, 0.16] | .15 | < .001 |
| Condition (control = 0; threat = 1) | | | | | 0.05 | [-0.06, 0.16] | .03 | .377 | 0.01 | [-0.10, 0.13] | .01 | .804 |
| General conspiracy beliefs | | | | | | | | | 0.03 | [-0.05, 0.11] | .04 | .439 |
| Food industry conspiracy beliefs | | | | | | | | | 0.13 | [0.05, 0.20] | .14 | .002 |
| $R^2$ | .10 | | | | .10 | | | | .13 | | | |
| $\Delta R^2$ | | | | | .001 | | | | .03* | | | |
| F | $F(12, 752) = 6.99^*$ | | | | $F(13, 751) = 6.51^*$ | | | | $F(15, 749) = 7.24^*$ | | | |

$^*p < .001$

positively related to conscious food choices. In Step 2, we introduced variable coding experimental condition (threat vs. control). We found that the effect of the experimental condition was non-significant. The effects of gender, age, shopping frequency, and subjective financial situation remained the same. In Step 3, we introduced food industry conspiracy beliefs and general conspiracy beliefs. We found that food industry conspiracy beliefs were a significant and positive predictor of conscious food choices, while the effects of the condition (threat vs. control) and of general conspiracy beliefs were non-significant. Effects of gender, age, shopping frequency, and subjective financial situation on the dependent variable remained significant.

In order to perform a full test of our hypotheses, we conducted a mediation analysis using model 4 in Process 3.5 [57]. Significance was tested with bootstrapped 95% confidence intervals for the unstandardized indirect effects, constructed with 10,000 resamples. The analysis, displayed in Fig 2, examined whether food industry conspiracy beliefs mediated the path between the experimental condition (threat vs. control) and conscious food choices. As covariates we used general conspiracy beliefs, gender, age, education level, settlement size, shopping frequency, and subjective financial situation. We found that the experimental condition positively and significantly predicted food industry conspiracy beliefs, $B = 0.19$, $SE = 0.05$, 95% $CI$ [0.09, 0.29], $p < .001$ and that, in turn, food industry conspiracy beliefs positively and significantly predicted conscious food choices, $B = 0.13$, $SE = 0.04$, 95% $CI$ [0.05, 0.20], $p = .002$. The indirect effect of the experimental condition on conscious food choices via food industry

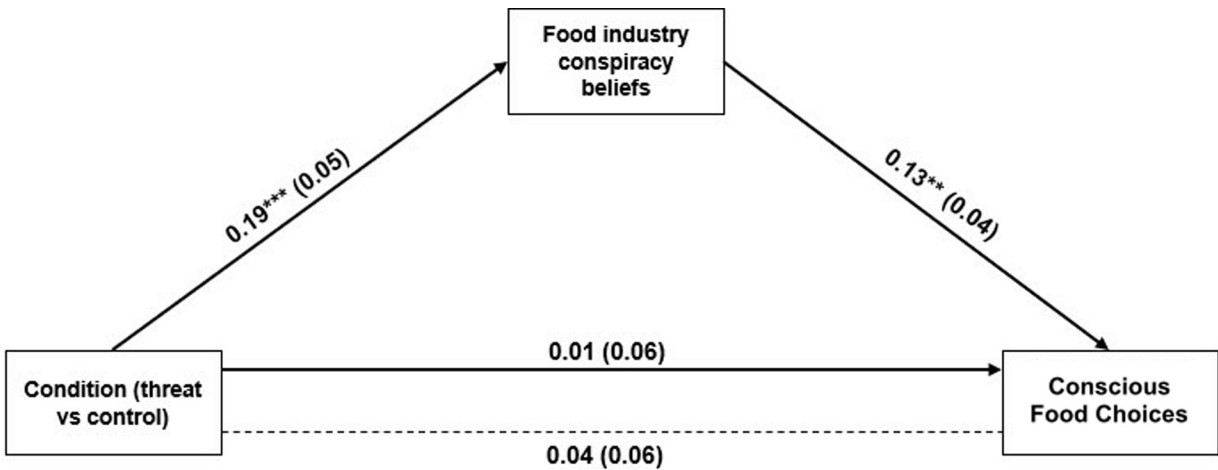

**Fig 2. Indirect effect of condition (threat vs. control) on conscious food choices via food industry conspiracy beliefs (Study 2).** Entries are unstandardized coefficients. Dotted line indicates total effect (not controlling for the third variable). $^*p < .05.$ $^{**}p < .01.$ $^{***}p < .001.$

conspiracy beliefs was positive and significant, $B = 0.02$, $SE = 0.01$, 95% $CI$ [0.01, 0.05]. All effects remained the same when we computed these analyses without the covariates. Next, we conducted similar analyses with general conspiracy beliefs as a mediator: the experimental condition did not predict general conspiracy beliefs significantly, $B = -0.05$, $SE = 0.05$, 95% $CI$ [-0.15, 0.06], $p = .372$, and general conspiracy beliefs was not a significant predictor of conscious food choices, $B = 0.03$, $SE = 0.04$, 95% $CI$ [-0.05, 0.10], $p = .439$. The indirect effect of the experimental condition on conscious food choices via general conspiracy beliefs was also non-significant, $B = -0.001$, $SE = 0.003$, 95% $CI$ [-0.01, 0.004].

In Study 2, we managed to replicate the pattern of results obtained in Study 1 with the use of better measurement tools. Specifically, we found a positive correlation between food industry conspiracy beliefs and conscious food choices. We also replicated the effect of age, suggesting that older individuals pay more attention to purchasing food consciously. However, we did not find a similar result for education. Moreover, Study 2 showed that inducing feelings of threat related to the possibility of purchasing food contaminated by a harmful bacteria (vs. control condition) increased food industry conspiracy beliefs, which were further positively correlated with conscious food choices.

In such a way, we found that inducing feelings of threat may indirectly strengthen adaptive consumer choices related to food purchasing behaviors via boosting context-specific conspiracy beliefs. In line with our predictions, these effects were not present in the context of generic conspiracist ideation, which did not increase after threat induction. Moreover, after accounting for the shared variance between food industry conspiracy beliefs and generic conspiracy ideation, only the former was found to predict conscious food choices.

## General discussion

In two studies, we investigated the phenomenon of food industry conspiracy beliefs. Using a nationally representative sample, in Study 1 we established that a third of Poles endorsed this form of conspiracy beliefs. Previous studies showed that belief in conspiracy theories was usually associated with maladaptive individual and societal outcomes [18]. However, we argued that there were some exceptions to this rule and showed (Study 1 and Study 2) that in some cases conspiracy beliefs were in fact related to adaptive behaviors. Specifically, we showed that those who endorsed food industry conspiracy beliefs were found to be more conscious

consumers (i.e., scored higher on conscious food choices). This seems to be in line with previous theorizing (e.g., [31]) emphasizing the mobilizing aspects of conspiracy beliefs and the role of higher suspicion and sensitivity to clues of danger that could decrease the possibility of missing insecure stimuli [35]. Additionally, it can be observed that while inducing context-specific threat (related to food content) increased the level of food industry conspiracy beliefs, it did not lead to a higher level of general conspiracy beliefs (Study 2). This evidence is yet another argument that food industry conspiracy beliefs, as measured in the present research, are qualitatively different from other, previously studied examples of conspiracy theories.

Importantly, in Study 2, we additionally demonstrated that a mere induction of feelings of threat (i.e., an article about a dangerous bacteria in food) did not directly change the consumers' perspective on conscious food choices. Rather, our analysis indicated that threat induction directly increased only food industry conspiracy beliefs, which were further positively linked to conscious food choices. This is consistent with previous research showing that people endorse conspiracy theories particularly when they experience feelings of anxiety or uncertainty [58, 59]. According to Van Prooijen [58], feelings of threat fuel a sense-making process focused on finding alleged enemies who can be blamed for unpleasant psychological states. In such a way, conspiracy theories offer structured maps of meaning and give simple explanations for uncertain situations [59, 60]. They help to track the enemy responsible for a threatening situation (e.g., food industry companies). This process, however, does not lead to threat reduction, but instead, seems to exaggerate the danger [51, 58]: we feel threatened, we have enemies so we should be careful and ready to fight. Previous researchers analyzed this mechanism from the perspective of intergroup relations (see [30]). On the one hand, they emphasized positive links between conspiracy beliefs and maladaptive intergroup outcomes (e.g., out-group hostility in times of peace; [28, 61]), but on the other, they elaborated on the evolutionary value of conspiracy theories that have been able to instill fear and anger in perceivers in times of war [58]. Our research extended this work by showing that conspiracy beliefs may also lead to adaptive intraindividual outcomes (i.e., paying more attention to the food products we choose).

Future research would do well to test what type of conspiracy beliefs may evoke adaptive behaviors (vs. be associated only with maladaptive ones). It is possible that only such beliefs that are based on a real threat (e.g., food contamination) may in some cases be related to positive outcomes. Potentially fertile ground for future research would also be to investigate the possible maladaptive concomitants of food industry conspiracy beliefs. In fact, it is possible that an obsessive focus on this type of convictions could also evoke undesirable psychological effects in the long term (e.g., lead to eating disorders such as orthorexia nervosa; [62]). One of the limitations was low (but acceptable; [63]) average variance explained for food industry conspiracy beliefs and conscious food choices. These scales should be psychometrically tested and revised in the future research. Future research is also needed to better investigate other possible predictors of food industry conspiracy beliefs as well as conscious food choices. For example, it would be interesting to check whether variables usually linked to conspiracy beliefs (e.g., need for cognitive closure; [59] or defensive self-evaluation; [64] would serve as significant predictors of conspiracy thinking also in this case. Other limitation was the decision about the order of the scales. In Study 1, we decided that the order of the scales should be rotated to maximize the validity of the research. In Study 2, we decided that food industry conspiracy beliefs should be presented before conscious food choices, as its possible underpinning. Although the variables were positively related to each other in both cases, future research would do well to further explore the potential influence the order of these scales might have on the results. Additionally, data measuring conscious food choices relied on self-reported declarations, so verifying whether a similar increase would be noted in actual shopping behavior is

needed. Given the findings of past research on consumer choices and implicit attitudes [7, 8], social and intangible attributes [65], as well as self-reported shopping behavior [66], we assume that the pattern of results obtained in the present studies would remain similar, though this would have to be verified in the future.

Also, future research would do well to better establish the causality of the observed relationships, for example, by experimentally manipulating the levels of food industry conspiracy beliefs. According to our predictions, boosting food industry conspiracy beliefs should lead to conscious food choices. On the other hand, we cannot exclude the possibility, that boosting conscious food choices may change the levels of food industry conspiracy beliefs. Future research is also needed to understand the influence of food industry conspiracy beliefs on consumer choices in a real shopping setting, with real products that can be inspected by the consumer before making a purchase decision. For example, it would be interesting to find out which aspects of conscious food choices (e.g., reading the nutrition information displayed on the label vs. using an app that with tell which food products are healthy) would be most popular among the consumers and which could be most effectively strengthened by food industry conspiracy beliefs. These issues require further empirical investigation. Similarly, given the novel character of the food industry conspiracy beliefs and the conscious food choices measurement tools, it would be beneficial to continue exploring the psychometric properties of these scales and to verify whether they would replicate in different cultural and economic contexts. One more issue was low R-squared in both Study 1 and Study 2 regression analyses [67]. These findings should be treated with caution and future studies should further analyze psychological concomitants of food industry conspiracy beliefs and conscious food choices.

The present research bears significant practical implications, as it points towards a psychological mechanism responsible for an increased willingness to pay more attention to the composition of purchased foods. Therefore, it could also be considered from the perspective of the broader concept of food integrity [68], which includes legal, moral, and ethical dimensions pertaining to the food supply and demand network. Identifying an efficient way of convincing individuals of the benefits of responsible consumption has been a burning issue in the last decades, especially given the general concern with global sustainability [69].

Importantly, although priming food-related threat may be a way to boost food industry conspiracy beliefs and, thus, increase conscious food choices, one should be aware of its potential shortcomings. In fact, previous research showed that feelings of threat [70] as well as conspiracy beliefs have negative consequences (e.g., lack of trust to government; [71] or antisemitic behaviours; [72]). Thus, one should remain cautious when employing this type of interventions. Still, materials elaborated for the purpose of this research could be analyzed by different entities, from local collectives or schools to international organizations, engaged in projects aimed at increasing people's awareness regarding the implications of conscious food choices. This seems particularly relevant in times when population obesity is accompanied by enormous food waste.

Overall, the current results allowed us to understand the role of food industry conspiracy beliefs in shaping conscious consumer choices. We showed that by increasing the level of this particular conspiracy belief through context-specific, food-related threat, individuals may become more susceptible to cues of danger and show greater readiness to reconsider their food purchasing decisions. Importantly, our research demonstrated that food industry conspiracy beliefs differed from the general notions of conspiracy studied before. The novel approach to the topic of conspiracies adopted in this research not only paves the way for a practical application of its results, but also points towards a yet unexplored area of study related to conspiracy beliefs, that is their possible adaptive outcomes.

## Author Contributions

**Conceptualization:** Marta Marchlewska.

**Data curation:** Marta Marchlewska, Zuzanna Molenda.

**Formal analysis:** Marta Marchlewska, Zuzanna Molenda, Marta Rogoza.

**Funding acquisition:** Marta Marchlewska.

**Investigation:** Marta Marchlewska, Dagmara Szczepańska, Adam Karakula, Zuzanna Molenda.

**Methodology:** Marta Marchlewska, Dagmara Szczepańska.

**Project administration:** Marta Marchlewska.

**Resources:** Marta Marchlewska.

**Supervision:** Marta Marchlewska.

**Visualization:** Adam Karakula, Marta Rogoza.

**Writing – original draft:** Marta Marchlewska, Dagmara Szczepańska, Adam Karakula.

**Writing – review & editing:** Marta Marchlewska, Dagmara Szczepańska, Adam Karakula, Zuzanna Molenda, Dominika Maison.

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
