## [Decision Letter · Decision Letter 0]

14 Mar 2022

PONE-D-22-05042The Devil is not as Black as He is Painted? On the Positive Relationship Between Food Industry Conspiracy Beliefs and Conscious Food ChoicesPLOS ONE

Dear Dr. Marchlewska,

Thank you for submitting your manuscript to PLOS ONE. After careful consideration, we feel that it has merit but does not fully meet PLOS ONE’s publication criteria as it currently stands. Therefore, we invite you to submit a revised version of the manuscript that addresses the points raised during the review process.

We look forward to receiving your revised manuscript.

Kind regards,

Hans De Steur

Academic Editor

PLOS ONE

Journal Requirements:

"This research was funded by National Science Centre Poland under Opus grant(2019/35/B/HS6/00123)."

"This research was funded by National Science Centre Poland under Opus grant  (2019/35/B/HS6/00123) awarded to MM. The funders had no role in study design, data collection and analysis, decision to publish, or preparation of the manuscript."

Reviewers' comments:

Reviewer's Responses to Questions

**Comments to the Author**

1. Is the manuscript technically sound, and do the data support the conclusions?

Reviewer #1: Yes

Reviewer #2: Partly

2. Has the statistical analysis been performed appropriately and rigorously? 

Reviewer #1: Yes

Reviewer #2: No

3. Have the authors made all data underlying the findings in their manuscript fully available?

Reviewer #1: Yes

Reviewer #2: Yes

4. Is the manuscript presented in an intelligible fashion and written in standard English?

Reviewer #1: Yes

Reviewer #2: Yes

5. Review Comments to the Author

Reviewer #1: 1. This paper is technically sound and consistant, approaching the research question with proper quantitative methods to demonstrate that food industry conspiracy belief is positively associated with consumers' conscious food choice.

2. The data analysis (Exploratory factor analysis and hierarchical regression using SPSS) is conducted properly and the procedure is well described. One recommendation is to refer to the communalities in the text, especially for study 2 which shows rather low (although still acceptable) average variance explained for food industry conspiracy beliefs and conscious food choices, to clarify whether all items are included in the construct.

3. The questionnaire, treatment and SPSS codes are available from the link on page 8. Full question items for general conspiracy beliefs is missing from the text and the shared data, and it would be better to be made available as well.

4. The quality of English is good enough.

As an additional comment, the practical implication(manuscript p.28) is not very clear, and it can be more specific about how the results should be interpreted and utilized. One concern is, boosting the level of food industry conspiracy means increasing consumers' distrust in or hostile view toward food companies, which is not generally desirable. While the association deserves investigation, I suppose that food industry conspiracy itself is not a "tool" to raise consumers' consciousness about food. This point should be acknowledged if the authors consider the same.

Reviewer #2: General comments:

The authors present the results from two studies with Polish consumers on their conspiracy beliefs about the food industry and how this relates to their conscious food choices. It is an interesting study field because, as the idiom in the title suggests, conspiracy believers are usually associated with maladaptive behaviour. Through both studies the researchers found a positive link between food industry conspiracy beliefs and conscious food choices.

Overall, the structure of the manuscript needs to be improved. Several of the specific comments below discuss some of the gaps or overlaps because of the structure. Subdividing study 1 and study 2 at the highest level creates repetition. More importantly, the fact that 2 constructs were measured in a different way in study 2 but still have the same name makes the interpretation of the results more difficult for the reader. The authors should consider a small change in the name of the variables. The change of measurement tools is not discussed in much detail. The reason behind this choice and the impact on the results should be addressed.

The first issue with the design of the study lies within the construct ‘conscious food choices’ and how it is interpreted. Not much information is provided on how the long version was developed, however for the short version the authors refer to a number of papers on socially responsible consumption. The items used to measure conscious food choices all refer to ‘I will pay attention to …’, which measures how informed consumers are when making their food choices. I would like the authors to add a discussion on why they assume adaptive behaviour based on this construct.

The second issue is the use of education and settlement size variables as dependent variables in the regression analysis. The authors do not mention any recoding of these variables so assuming they used the data as is, this is a wrong approach because these are not interval data.

Specific comments:

100: Throughout the manuscript it seems the authors only consider the food conspiracy beliefs in relation to food safety issues. Considering the definition of food conspiracy beliefs on line 128, I believe the broader concept of food integrity could be used here, which includes food safety but also authenticity. Irregularities with food products’ integrity do not only apply to food safety issues but can be food fraud or mislabelling (f.e. organic or country of origin labelling).

111: The authors should include at least one example from Europe or Poland.

136: The last sentence of this paragraph is unclear

191: It would be interesting to discuss the order of the questions in the questionnaire. Were the conspiracy beliefs asked before the conscious food choices, and if so why did the researches choose this order and how could it have affected the results?

204: The authors report a reliability measure for 4 items of the scale ‘food conspiracy beliefs’. Later on in the manuscript it is mentioned that this scale was measured using 4 items in study 1 and 14 items in study 2. It is important to make this difference clear in the methods section.

244: The authors chose to discuss both studies separately and make a ‘Results and discussion’ section for each of the studies. There is actually not much discussion of the results in that section, and there is another section ‘General discussion’ later on. I would suggest to restructure and rename the sections of the manuscript to make it easier to navigate as a reader and avoid repetition. Overall for both study 1 and study 2 I would be interested in more discussion of the results.

260: Table 1 shows the correlation matrix, including education and settlement size as variables. Can you provide more details about how these were measured and how they could be considered continuous variables?

268: Significant relationship between age and education and conscious food choices are found. Is this in line with your expectations?

274: Please discuss the R squared value of the regression model

274: Unclear what you mean with term ‘basic prediction’

291: general instead of generic

326: Explain in more detail how the items were designed. On line 298, the authors mention that the conceptual principles to develop the tools in study 1 still applied. This made me assume that the added items would follow the principles of line 193, being ‘Each item included three elements: an implied agent (1) secretly undertaking specific action (2) to obtain some type of gain (3). However, when reading the added items, this seems not to be the case.

348: In similar vein, explain how the nine items for the conscious food choices were developed. Overall, the use of the same construct names ‘food industry conspiracy beliefs’ and ‘conscious food choices’ throughout the paper, while measuring them differently is confusing. The authors should differentiate clearly.

370: Have the authors considered if they can accept participants that shop for groceries once a month or less as valid respondents for this study?

370: Was the data analysis carried out with the scale of the variable ‘frequency grocery shopping’ as such, or was this recoded to actual frequency?

381: The researches could elaborate more detailed about why they chose to use mediation analysis and how assumptions for mediation analysis were checked.

383: avoid the use of ‘basic variables’

385: use ‘correlation’ instead of ‘link’, and discuss here that this was not significant in study 1. Could the difference be related to the new way of measuring conscious food choice?

388: since including shopping frequency, I suggest using the term socio-demographic instead of demographic

388: Later on in the results, a significant effect of the treatment (threat vs control) is found. I would expect an explanation why you discuss the difference in socio-demographics without differentiating between those treatment groups. As opposed to study 1, there is difference in conscious food choices between males and females in study 2. Would the difference results be due to different way of measuring conscious food choices or because this study included the experimental manipulation?

392: Age is positively correlated to food conspiracy beliefs but negatively to general conspiracy beliefs, this is an interesting finding. Discuss by comparing to previous studies.

405: avoid the term items but use variables

414: Authors should discuss and interpret R squared of the final model.

434: Please rephrase ‘a significant negative effect of gender’; specify that you entered gender as a dummy variable

445: In study 1, education was significant, but in study 2 it turned out not to be. This difference should be discussed.

477: This paragraph is generalizing the results too much. There were difference between the results of study 1 and study 2 and they are not sufficiently reported and discussed.

486: The authors always refer to the number of respondents that endorse food conspiracy beliefs, based on the results of study 1. In study 2 they used better measurement for this variable, so it seems contradictory that they don’t use the results from the better measurement to give the reader an indication of the food conspiracy beliefs of Polish consumers. The ‘threat’ group could have been influenced by the message, however the control group was not. It would also be interesting to report the difference in mean value of the food conspiracy beliefs for the 2 treatment groups.

6. PLOS authors have the option to publish the peer review history of their article (what does this mean?). If published, this will include your full peer review and any attached files.

Reviewer #1: No

Reviewer #2: No

---

## [Author Response · Author response to Decision Letter 0]

28 Apr 2022

Please note that all the answers could be found in the uploaded file (Plos rev Letter. 27.04 FINAL.docx). They could be also found below.

April 27th, 2022

RE: Notification of decision: PONE-D-22-05042

Dear Professor Hans De Steur,

Thank you for your invitation to revise and resubmit our manuscript entitled “The devil is not as black as he is painted? On the positive relationship between food industry conspiracy beliefs and conscious food choices'' (PONE-D-22-05042). We are very grateful to you and the reviewers for the responses. In this letter, we outline the changes we have made to the paper regarding the minor revision that was requested. Also, we would like to kindly ask you to include our funding statement in the Acknowledgments section (final version of the manuscript) as this is explicitly required by our founding source (National Science Centre). 

Thank you in advance! 

Authors

Editor’s comments

E.1.1 Please include your full ethics statement in the ‘Methods’ section of your manuscript file. In your statement, please include the full name of the IRB or ethics committee who approved or waived your study, as well as whether or not you obtained informed written or verbal consent. If consent was waived for your study, please include this information in your statement as w well. 

Thank you for this suggestion, we have now included an appropriate ethics statement on page 8, which now read as follows: “Both studies were conducted in accordance with the Declaration of Helsinki and approved by the Research Ethics Committee of the Institute of Psychology, Polish Academy of Sciences (number of approval: 26/X/2020). Informed written consent was obtained from all subjects involved in the study.”

Reviewer #1

R1.1. This paper is technically sound and consistant, approaching the research question with proper quantitative methods to demonstrate that food industry conspiracy belief is positively associated with consumers' conscious food choice.

We thank reviewer for their positive comments. 

R1.2. The data analysis (Exploratory factor analysis and hierarchical regression using SPSS) is conducted properly and the procedure is well described. One recommendation is to refer to the communalities in the text, especially for study 2 which shows rather low (although still acceptable) for food industry conspiracy beliefs and conscious food choices, to clarify whether all items are included in the construct.

Thank you for pointing this out. We have now elaborated on this issue in the discussion section. Page 28 now reads: 

“One of the limitations was low (but acceptable; [63]) average variance explained for food industry conspiracy beliefs and conscious food choices. These scales should be psychometrically tested and revised in the future research. Future research is also needed to better investigate other possible predictors of food industry conspiracy beliefs as well as conscious food choices. For example, it would be interesting to check whether variables usually linked to conspiracy beliefs (e.g., need for cognitive closure; [64] or defensive self-evaluation; [65] would serve as significant predictors of conspiracy thinking also in this case.” 

R.1.2 The questionnaire, treatment and SPSS codes are available from the link on page 8. Full question items for general conspiracy beliefs is missing from the text and the shared data, and it would be better to be made available as well.

Thank you for pointing this out! General conspiracy beliefs scale is now available in the OSF folder, available through the link on page 8: https://osf.io/h4x5v/?view_only=d7fd112e8bcb470eb9ec4090882d0e2b

R.1.3 As an additional comment, the practical implication (manuscript p.28) is not very clear, and it can be more specific about how the results should be interpreted and utilized. One concern is, boosting the level of food industry conspiracy means increasing consumers' distrust in or hostile view toward food companies, which is not generally desirable. While the association deserves investigation, I suppose that food industry conspiracy itself is not a "tool" to raise consumers' consciousness about food. This point should be acknowledged if the authors consider the same.

Thank you for that comment. We agree with that, and we have now explained that concern in the discussion section.

Page 30 now reads: “Importantly, although priming food-related threat may be a way to boost food industry conspiracy beliefs and, thus, increase conscious food choices, one should be aware of its potential shortcomings. In fact, previous research showed that feelings of threat [71] as well as conspiracy beliefs have negative consequences (e.g., lack of trust to government; [72] or antisemitic behaviours; [73]). Thus, one should remain cautious when employing this type of interventions.” 

Reviewer #2

R2.1. The authors present the results from two studies with Polish consumers on their conspiracy beliefs about the food industry and how this relates to their conscious food choices. It is an interesting study field because, as the idiom in the title suggests, conspiracy believers are usually associated with maladaptive behaviour. Through both studies the researchers found a positive link between food industry conspiracy beliefs and conscious food choices. 

We thank the reviewer for their positive comments.

R2.2. Overall, the structure of the manuscript needs to be improved. Several of the specific comments below discuss some of the gaps or overlaps because of the structure. Subdividing study 1 and study 2 at the highest level creates repetition. More importantly, the fact that 2 constructs were measured in a different way in study 2 but still have the same name makes the interpretation of the results more difficult for the reader. The authors should consider a small change in the name of the variables. The change of measurement tools is not discussed in much detail. The reason behind this choice and the impact on the results should be addressed.

Thank you for your comment. We decided to subdivide Study 1 and 2 as they had different designs (i.e., Study 1 was cross-sectional, Study 2 was experimental). We believed they should not be merged to avoid confusion. We also changed the names of the scales in the method section by adding “short scale” (pp. 9-10) and “full scale” (pp. 15-17) to the previous measures’ labels. We used longer versions of our measurement tools to analyze the full spectrum of the variables of our main interest (i.e., conspiracy beliefs and conscious food choices). This is now clarified on page 14:

“One limitation of Study 1 was that we measured the crucial variables (e.g., food industry conspiracy beliefs and conscious food choices) with the use of short (four- and three-item) scales. Therefore, in Study 2 we examined whether the pattern of results obtained in Study 1 would conceptually replicate if we used better measurement tools. We operationalized the food industry conspiracy beliefs and conscious food choices with 14 and 9 items respectively. The conceptual principles applied while developing the tools remained the same as in Study 1, with some items including an implicit allusion to the action secretly undertaken by food industry companies for their own benefit. Still, both extended versions of the scales showed high reliability (listed below) and Exploratory Factor Analysis provided single factor solutions for both of them.” 

In such a way, we extended the scales used in Study 1 with additional items referring to, for example, adding harmful, addictive substances (in the food industry conspiracy beliefs scale) and buying healthy food in local, trusted places (in the conscious food choices scale). 

Moreover, we conducted additional analyses to find out whether the pattern of results would be similar when considering only the same items as in Study 1 (short version). The results remained the same (please find the details below and in the OSF folder). 

Mediation Analyses Using Same Items as in Study 1 (Study 2)

In order to perform a full test of our hypotheses, we conducted a mediation analysis using model 4 in Process 3.5 [57]. Significance was tested with bootstrapped 95% confidence intervals for the unstandardized indirect effects, constructed with 10,000 resamples. The analysis, displayed in Fig 2, examined whether food industry conspiracy beliefs mediated the path between the experimental condition (threat vs. control) and conscious food choices. As covariates we used general conspiracy beliefs, gender, age, education level, settlement size, shopping frequency, and subjective financial situation. We found that the experimental condition positively and significantly predicted food industry conspiracy beliefs, B = 0.18, SE = 0.06, 95% CI [0.07, 0.29], p = .002 and that, in turn, food industry conspiracy beliefs positively and significantly predicted conscious food choices, B = 0.10, SE = 0.04, 95% CI [0.01, 0.18], p = .024. The indirect effect of the experimental condition on conscious food choices via food industry conspiracy beliefs was positive and significant, B = 0.02, SE = 0.01, 95% CI [0.001, 0.041]. All effects remained the same when we computed these analyses without the covariates. Next, we conducted similar analyses with general conspiracy beliefs as a mediator: the experimental condition did not predict general conspiracy beliefs significantly, B = -0.03, SE = 0.06, 95% CI [-0.14, 0.08], p = .550, and general conspiracy beliefs was not a significant predictor of conscious food choices, B = 0.04, SE = 0.04, 95% CI [-0.05, 0.13], p = .364. The indirect effect of the experimental condition on conscious food choices via general conspiracy beliefs was also non-significant, B = -0.001, SE = 0.004, 95% CI [-0.010, 0.005].

R2.3. The first issue with the design of the study lies within the construct ‘conscious food choices’ and how it is interpreted. Not much information is provided on how the long version was developed, however for the short version the authors refer to a number of papers on socially responsible consumption. The items used to measure conscious food choices all refer to ‘I will pay attention to …’, which measures how informed consumers are when making their food choices. I would like the authors to add a discussion on why they assume adaptive behaviour based on this construct.

Thank you for drawing attention to this issue. Indeed, the scale developed for the purpose of this research was inspired by past work on responsible consumption and it was our intention to use examples of behaviours that had been identified as such, for example checking the product’s ingredients, country of origin, or degree of processing. Still, while socially responsible consumption "can promote social causes consumers deem important" (Francois-Lecompte & Roberts, 2006, p. 51), in our studies, we were not interested in the social/environmental/political motives of food choices. Our focus was more on the individual's health. It seems that socially responsible consumption is a broader term than "conscious food choices." We believe that these constructs are related, but this needs further empirical investigation. In the case of adaptiveness, we argue that conscious food choices - made with an awareness and sensitivity to clues of potential danger or some kind of risks - are adaptive in the sense that they can prevent individuals from jeopardizing their health. Maladaptive, on the other hand, would be exposing yourself to the dangers of potential contamination of some product. In this sense, maladaptive means thoughtless, inconsiderate, which stands the opposite of "conscious choice."

The extended version of the scale (used in Study 2) was still based on the same principles as in Study 1, though it was extended to check if the results would replicate when we used a longer tool, hence it also included a wider variety of examples conscious shopping behaviours. Still, both scales formed a single factor in the Exploratory Factor Analysis. The phrase "I will pay attention to" we often used in our items is directly related to the term "conscious choices." In this way, we aimed to emphasize that food choices are made with awareness and attention and, thus, are conscious. Of course, in this case, we are aware that this measurement is only declarative and we acknowledge that it is a limitation and mention that in the the paragraph describing future directions. We see that the scale could be interpreted as reflecting how much the consumers are informed about the composition of the products they consider buying and believe it could translate into adaptive behavior for several reasons. First, past research on social and other intangible product attributes demonstrated that they did in fact influence actual product choice (Auger et al., 2010). Second, self-reported shopping behavior has been previously associated with actual decisions, though they dependent on the type of product (Moser, 2016). Third, research on implicit vs explicit attitudes towards brands showed that they were in fact positively associated [7, 8]. Based on these findings, we assume that the conscious food choices scale we developed would be positively associated with specific actions too, though it would have to be checked. We have now added a clearer explanation regarding that assumption in the Discussion section on p. 28-29, which reads:

“Additionally, data measuring conscious food choices relied on self-reported declarations, so verifying whether a similar increase would be noted in actual shopping behaviour is needed. Given the findings of past research on consumer choices and implicit attitudes [7, 8], social and intangible attributes [66], as well as self-reported shopping behaviour [67], we assume that the pattern of results obtained in the present studies would remain similar, though this would have to be verified in the future.”

R2.3. The second issue is the use of education and settlement size variables as dependent variables in the regression analysis. The authors do not mention any recoding of these variables so assuming they used the data as is, this is a wrong approach because these are not interval data.

Thank you. We used demographics as independent variables to check for their potential role in shaping food industry conspiracy beliefs and conscious food choices and, especially, to find out whether the pattern of results would stay the same after controlling for age, gender etc. These were not the variables of our main interest, but still potentially interesting. Importantly, when conducted analyses without these variables, the results remained the same. 

Specific comments:

100: Throughout the manuscript it seems the authors only consider the food conspiracy beliefs in relation to food safety issues. Considering the definition of food conspiracy beliefs on line 128, I believe the broader concept of food integrity could be used here, which includes food safety but also authenticity. Irregularities with food products’ integrity do not only apply to food safety issues but can be food fraud or mislabelling (f.e. organic or country of origin labelling).

Thank you for your comment. We appreciate your suggestion, however, if we were to replace the concept of food conspiracy theories with food integrity, the entire theoretical background on which we base the main hypothesis would have to be different. The scale was developed to reflect the fundamental principles of conspiracy thinking, namely that some secret agents are plotting behind our back to obtain some kind of a gain and that the results of that secret action may be potentially threatening to us (Douglas et al., 2019). As far as we are aware, food integrity is a much broader concept including many specific issues, but it does not contain that element of conspiracy thinking we employed in the scale. Of course, it is an interesting insight to consider the results of our research from the perspective of food integrity and we have now elaborated on this idea in the discussion section on pages 30: 

“The present research bears significant practical implications, as it points towards a psychological mechanism responsible for an increased willingness to pay more attention to the composition of purchased foods. Therefore, it could also be considered from the perspective of the broader concept of food integrity [69], which includes legal, moral, and ethical dimensions pertaining to the food supply and demand network. Identifying an efficient way of convincing individuals of the benefits of responsible consumption has been a burning issue in the last decades, especially given the general concern with global sustainability [70].”

111: The authors should include at least one example from Europe or Poland.

Thank you for that comment. We added some statistics from the European Union.

Page 5 now reads: “Another example is the European Union, where more than 90,000 cases of Salmonella are recorded each year and the main risk of infection in humans is associated with the consumption of contaminated food [43].”

136: The last sentence of this paragraph is unclear

Thank you for your comment! We have now rephrased this sentence to improve its clarity.

Page 6 now reads: “However, it needs to be highlighted that our intention was not to verify the validity of these accusations, but to explore the psychological concomitants of conspiracy beliefs related to the food industry. “ 

191: It would be interesting to discuss the order of the questions in the questionnaire. Were the conspiracy beliefs asked before the conscious food choices, and if so why did the researches choose this order and how could it have affected the results?

Thank you for your comment. Indeed, this issue requires explanation. Thus, we elaborated on it in the discussion section.

Page 28 now reads:

“Other limitation was the decision about the order of the scales. In Study 1, we decided that the order of the scales should be rotated to maximize the validity of the research. In Study 2, we decided that food industry conspiracy beliefs should be presented before conscious food choices, as its possible underpinning. Although the variables were positively related to each other in both cases, future research would do well to further explore the potential influence the order of these scales might have on the results.”

204: The authors report a reliability measure for 4 items of the scale ‘food conspiracy beliefs’. Later on in the manuscript it is mentioned that this scale was measured using 4 items in study 1 and 14 items in study 2. It is important to make this difference clear in the methods section.

Thank you for that comment. Scales used in Study 2 are extended versions of the measurement tools employed in Study 1. For the sake of clarity, we changed the names of the scales in the method section by adding “short scale” (pp. 9-10) and “full scale” (pp. 15-17) to the previous measures’ labels. It is described in the methods on pages 9 and 16 

244: The authors chose to discuss both studies separately and make a ‘Results and discussion’ section for each of the studies. There is actually not much discussion of the results in that section, and there is another section ‘General discussion’ later on. I would suggest to restructure and rename the sections of the manuscript to make it easier to navigate as a reader and avoid repetition. Overall for both study 1 and study 2 I would be interested in more discussion of the results.

Thank you for that comment. The purpose of short discussion after each study’s results is more to summarize the study and provide initial directions of interpretation. Detailed discussion is in the General discussion section, and it focuses on particular aspects of both studies. We would prefer not to rename the sections, because we believe that its current version is more in line with previous papers on conspiracy beliefs (e.g., Marchlewska et al., 2021, Jolley & Douglas, 2014, Stojanov et al., 2020, Alsuhibani et al., 2022). Naturally, we will change it if you think it is necessary.

Moreover, the revised version of the manuscript includes a more detailed discussion. Specifically, we elaborated more on the limitations and future directions issues.

260: Table 1 shows the correlation matrix, including education and settlement size as variables. Can you provide more details about how these were measured and how they could be considered continuous variables?

Thank you for this comment. We explain they way of measurement of these variables in details on page 10 in section called “Covariates”. We agree that these variables are not continuous in nature, thus, requiring different than Pearson correlation coefficients. Please note, however, that we tested these associations using Spearman rank-order coefficient, finding nearly identical results. Thus, to ease the presentation of the results, we left Pearson's correlations coefficients. We did, however, added an additional footnote emphasizing that the correlation to categorical variables (i.e., education and settlement size) were essentially the same when rank-order correlation coefficient was applied. 

Table 1 on page 12 now reads: Note. We also conducted correlation analyses using Spearman test. Results remained the same.

In order to be consistent we also provided footnote to Table 3.

Table 3 on page 20 now reads: Note. We also conducted correlation analyses using Spearman test. Results remained the same.

268: Significant relationship between age and education and conscious food choices are found. Is this in line with your expectations?

Thank you for this remark. Age and education were not our main variables of interest, we used them as covariates and we had no specific expectations related to demographic variables. However, as you noted, we found some interesting significant relationships and they should be reported in the discussion. Thus, we added new sentences to Study 1 and 2.

Page 13 now reads:

“Additionally, we found that higher levels of education and age predicted more conscious food choices. It seems that older, more life-experienced, and better-educated individuals focus more on conscious choices while purchasing food products.”

Page 26 now reads as follows:

“We also replicated the effect of age, suggesting that older individuals pay more attention to purchasing food consciously. However, we did not find a similar result for education.”

274: Please discuss the R squared value of the regression model

Thank you for that comment. We added further interpretation in the discussion.

p 29. now reads: “One more issue was low R-squared in both Study 1 and Study 2 regression analyses [68]. These findings should be treated with caution. Future studies should further analyze psychological concomitants of food industry conspiracy beliefs and conscious food choices.”

274: Unclear what you mean with term ‘basic prediction’

Thank you for that comment. The point of this phrase was to indicate predictions connected to our hypotheses, but to make it clearer we decided to delete word “basic”.

291: general instead of generic

Thank you for that comment. We have now changed the word accordingly.

326: Explain in more detail how the items were designed. On line 298, the authors mention that the conceptual principles to develop the tools in study 1 still applied. This made me assume that the added items would follow the principles of line 193, being ‘Each item included three elements: an implied agent (1) secretly undertaking specific action (2) to obtain some type of gain (3). However, when reading the added items, this seems not to be the case. 

Thank you for this comment. Indeed, the same principles still applied in Study 2, but some of the items included an implicit allusion to the core elements of conspiracy theories, rather than an explicit one [29]. For example, the item “Food processing companies use genetically modified ingredients without letting the consumers know” contains an element of implied gain because if information is withheld there must be a reason for doing that. Similarly, if “Nobody really knows what is inside of food products” it is because someone is purposely not telling us that. To make the issue clearer, we have now added the following to the section introducing Study 1 (p. 14):

“The conceptual principles applied while developing the tools remained the same as in Study 1, with some items including an implicit allusion to the action secretly undertaken by food industry companies for their own benefit. Still, both extended versions of the scales showed high reliability (listed below) and Exploratory Factor Analysis provided single factor solutions for both of them.” 

348: In similar vein, explain how the nine items for the conscious food choices were developed. Overall, the use of the same construct names ‘food industry conspiracy beliefs’ and ‘conscious food choices’ throughout the paper, while measuring them differently is confusing. The authors should differentiate clearly.

Thank you for this suggestion. We added “short scale” and “full scale” at the end of the scales’ name in methods section as suggested. We understand that using the same name can be confusing, however, both the short versions used in Study 1 and the long versions in Study 2 measure the same constructs and yield the same patterns of results. Scales used in Study 2 include the original items employed in Study 1. To demonstrate the validity of the construct we have now performed some additional analysis for Study 2 including only the original 4 items from Study 1. We included these analyses in OSF folder.

https://osf.io/h4x5v/?view_only=d7fd112e8bcb470eb9ec4090882d0e2b

370: Have the authors considered if they can accept participants that shop for groceries once a month or less as valid respondents for this study?

Thank you for that comment. Only 0.9% of respondents said that they went shopping once a month or never. However, we conducted additional analysis and found that the results did not change when we excluded these participants. 

370: Was the data analysis carried out with the scale of the variable ‘frequency grocery shopping’ as such, or was this recoded to actual frequency?

The frequency of grocery shopping was coded as it was presented within the scale. We did not recode this variable to reflect the actual frequency as we did not have had such objective data.

381: The researches could elaborate more detailed about why they chose to use mediation analysis and how assumptions for mediation analysis were checked.

Thank you for this comment. We used mediation to examine whether the indirect effect of experimental condition on conscious food choices via food industry conspiracy beliefs.

The relationship of condition and conscious food choices exists through food conspiracy beliefs.

We checked the normality of the variables, linearity of the model, homoscedasticity, autocorrelation of the variables, collinerality, measurement error and if errors have homogeneous variance. All of the results were acceptable enough to use mediation analysis.

383: avoid the use of ‘basic variables’

Thank you for that comment. We have now refrained from using this term.

385: use ‘correlation’ instead of ‘link’, and discuss here that this was not significant in study 1. Could the difference be related to the new way of measuring conscious food choice?

Thank you for that comment. We have now changed “link” to “correlation” when possible. Conscious food choices scale was used in full version in Study 2, which in our belief might have resulted in such an outcome. Indeed, the correlation between conscious food choices and general conspiracy beliefs (r = .04, p = .388) was not significant in Study 1, but it was significant (r = .09, p = .017) in Study 2. Moreover, we conducted additional correlation analysis, employing only the items used in Study 1 for the conscious food choices scale, and this relationship was also not significant (r = .07, p = .060). Thus, it might be the case that the use of a longer, more complex measurement of conscious food choices has resulted in such an outcome, as you suggested in this comment. 

388: since including shopping frequency, I suggest using the term socio-demographic instead of demographic

Thank you for that comment. We changed it where it was necessary. 

388: Later on in the results, a significant effect of the treatment (threat vs control) is found. I would expect an explanation why you discuss the difference in socio-demographics without differentiating between those treatment groups. As opposed to study 1, there is difference in conscious food choices between males and females in study 2. Would the difference results be due to different way of measuring conscious food choices or because this study included the experimental manipulation?

Thank you for that comment. We had no specific hypothesis on the different effects of threat on the variables of main interest (e.g., conspiracy beliefs) among male vs. female participants. In other words, according to our knowledge, there is no theoretical justification for such. 

For this reason, we would prefer to stick to the current version of the analytical section. However, we would be glad to add additional results in the supplementary file should you deem this appropriate.

392: Age is positively correlated to food conspiracy beliefs but negatively to general conspiracy beliefs, this is an interesting finding. Discuss by comparing to previous studies.

Thank you for that comment. This is an interesting finding, however, it only occurred in Study 2. On the contrary, Study 1 showed that age did not correlate significantly with food industry conspiracy beliefs or general conspiracy beliefs. Thus, with such inconsistencies between the studies, we would suggest treating these relationships between age, general conspiracy beliefs, and food industry conspiracy beliefs with caution, especially while the latter is the construct that – to our knowledge – has never been investigated before. It is possible that food industry conspiracy beliefs are characterized by some specificity related to age. For example, it is plausible that older individuals who have more experience with the food industry or with purchasing food over their lives may be more prone to believing in such theories. However, we do not have enough evidence to theorize about it.

We are certain that it requires more research, but we are not sure about discussing it in that manuscript. Of course, we would be happy to include a more detailed discussion of these issues in the manuscript, should you deem this appropriate

405: avoid the term items but use variables

Thank you for that comment. We changed it where it was necessary.

414: Authors should discuss and interpret R squared of the final model.

Thank you for that comment. We added interpretation in the general discussion section.

p 29. now reads: “One more issue was low R-squared in both Study 1 and Study 2 regression analyses [68]. These findings should be treated with caution. Future studies should further analyze psychological concomitants of food industry conspiracy beliefs and conscious food choices.”

434: Please rephrase ‘a significant negative effect of gender’; specify that you entered gender as a dummy variable 

Thank you for that comment. We rephrased that statement. 

Page 23 now reads: Gender was a significant and negative predictor of conscious food choices

We also added an information about the coding of this variable on page 10:

“In addition to age and gender (coded Female = 0, Male =1)”

445 : In study 1, education was significant, but in study 2 it turned out not to be. This difference should be discussed.

The zero-order relation between education to food industry conspiracy beliefs was indeed non-significant in Study 1 and significant in Study 2. We are sure that it requires more research and mentioned about it in discussion.

p. 29 now reads: “Another issue is that education was weakly and negatively related to food industry conspiracy beliefs (although this relation was non-significant in Study 1). This surely requires future research and better verification.”

477: This paragraph is generalizing the results too much. There were difference between the results of study 1 and study 2 and they are not sufficiently reported and discussed.

Thank you for that comment. We understand why this paragraph seemed overgeneralized. However, it is a short discussion, in which we aimed to summarize the results of Study 2 only. The results of both studies are precisely described and discussed in the General discussion. 

Things that are mentioned in short discussion are also included in general discussion, where they are more widely talked over. The example of verse 497 discussed in general discussion on p. 26-27:

“However, we argued that there were some exceptions to this rule and showed (Study 1 and Study 2) that in some cases conspiracy beliefs were in fact related to adaptive behaviors. Specifically, we showed that those who endorsed food industry conspiracy beliefs were found to be more conscious consumers (i.e., scored higher on conscious food choices).” 

In the General Discussion, we also highlighted some differences between Study 1 and Study 2. For example, page 28 says:

Other limitation was the decision about the order of the scales. In Study 1 we decided that the order of the scales should be rotated to maximize the validity of the research. In Study 2 we decided that food industry conspiracy beliefs should be presented before conscious food choices, as its possible underpinning. Although the variables were positively related to each other in both cases, future research would do well to further explore the potential influence the order of these scales might have on the results.

486: The authors always refer to the number of respondents that endorse food conspiracy beliefs, based on the results of study 1. In study 2 they used better measurement for this variable, so it seems contradictory that they don’t use the results from the better measurement to give the reader an indication of the food conspiracy beliefs of Polish consumers. The ‘threat’ group could have been influenced by the message, however the control group was not. It would also be interesting to report the difference in mean value of the food conspiracy beliefs for the 2 treatment groups.

Thank you for that comment. The whole sample was representative - with random assignment to conditions, but not in all groups. Thus, frequencies based on Study 2 would not be necessarily more reliable than those in Study 1. Due to the experimental manipulation conducted in Study 2, we restrained from adding frequencies for the whole sample (for those exposed to a threat material and those exposed to control material). 

However, we conducted these analyses using independent t-test which are in line with regression analyses reported in the manuscript. We present them below:

Independent t-test Analysis of Main Variables Between Threat and Control Group

Variable Threat group Control group t(763) p

 M SD M SD 

Food industry conspiracy beliefs 3.21 0.93 2.95 0.91 -3.85 <.001

Conscious food choices 3.62 0.80 3.54 0.82 -1.35 .178

General conspiracy beliefs 2.75 0.98 2.68 0.93 -1.04 .298

References:

Alsuhibani, A., Shevlin, M., Freeman, D., Sheaves, B., & Bentall, R. P. (2022). Why conspiracy theorists are not always paranoid: Conspiracy theories and paranoia form separate factors with distinct psychological predictors. PloS one, 17(4), e0259053.

Auger, P., Devinney, T. M., Louviere, J. J., & Burke, P. F. (2010). The importance of social product attributes in consumer purchasing decisions: A multi-country comparative study. International Business Review, 19(2), 140-159. 

Douglas, K. M., Uscinski, J. E., Sutton, R. M., Cichocka, A., Nefes, T., Ang, C. S., & Deravi, F. (2019). Understanding conspiracy theories. Political Psychology, 40, 3-35.

Francois-Lecompte, A., & Roberts, J. A. (2006). Developing a measure of socially responsible consumption in France. Marketing Management Journal, 16(2).

Jolley, D., & Douglas, K. M. (2014). The effects of anti-vaccine conspiracy theories on vaccination intentions. PloS one, 9(2), e89177.

Marchlewska, M., Górska, P., Malinowska, K., & Jarosław, K. (2021). Threatened masculinity: Gender-related collective narcissism predicts prejudice toward gay and lesbian people among heterosexual men in Poland. Journal of homosexuality, 1-16.

Moser, A. K. (2016). Consumers' purchasing decisions regarding environmentally friendly products: An empirical analysis of German consumers. Journal of Retailing and Consumer Services, 31, 389-397.

Stojanov, A., Bering, J. M., & Halberstadt, J. (2020). Does Perceived lack of control lead to conspiracy theory beliefs? Findings from an online MTurk sample. PloS one, 15(8), e0237771.

---

## [Decision Letter · Decision Letter 1]

20 Jun 2022

PONE-D-22-05042R1The Devil is not as Black as He is Painted? On the Positive Relationship Between Food Industry Conspiracy Beliefs and Conscious Food ChoicesPLOS ONE

Dear Dr. Marchlewska,

Thank you for submitting your manuscript to PLOS ONE. After careful consideration, we feel that it has merit but does not fully meet PLOS ONE’s publication criteria as it currently stands. Therefore, we invite you to submit a revised version of the manuscript that addresses the points raised during the review process.

We look forward to receiving your revised manuscript.

Kind regards,

Hans De Steur

Academic Editor

PLOS ONE

Additional Editor Comments:

Dear authors,

Given the concerns of one reviewer regarding the statistical analysis, I recommended major revision in order to allow you enough time to address these crucial concerns.

Reviewers' comments:

Reviewer's Responses to Questions

**Comments to the Author**

1. If the authors have adequately addressed your comments raised in a previous round of review and you feel that this manuscript is now acceptable for publication, you may indicate that here to bypass the “Comments to the Author” section, enter your conflict of interest statement in the “Confidential to Editor” section, and submit your "Accept" recommendation.

Reviewer #2: All comments have been addressed

2. Is the manuscript technically sound, and do the data support the conclusions?

Reviewer #2: Partly

3. Has the statistical analysis been performed appropriately and rigorously? 

Reviewer #2: No

4. Have the authors made all data underlying the findings in their manuscript fully available?

Reviewer #2: Yes

5. Is the manuscript presented in an intelligible fashion and written in standard English?

Reviewer #2: Yes

6. Review Comments to the Author

Reviewer #2: Thank you to the authors for there extensive answer to my comments on the original manuscript. You have made additions and changes that have improved the manuscript. Most of the comments have clarified some of the issues I had with the paper.

I would still hope you can further clarify why education and settlement sizee, used as covariates are treated as continuous variables.

7. PLOS authors have the option to publish the peer review history of their article (what does this mean?). If published, this will include your full peer review and any attached files.

Reviewer #2: No

---

## [Author Response · Author response to Decision Letter 1]

27 Jun 2022

Reviewer #2

Reviewer #2: Thank you to the authors for the extensive answer to my comments on the original manuscript. You have made additions and changes that have improved the manuscript. Most of the comments have clarified some of the issues I had with the paper.

I would still hope you can further clarify why education and settlement sizee, used as covariates are treated as continuous variables.

Once again, thank you for all your comments! We have now refrained from using education and settlement size as continuous variables and based on dummy coding procedure (Hutcheson & Sofroniou, 1999) that allowed us to use these variables in regression analyses. 

Methods section now reads:

Both education and settlement size were explanatory variables of categorical level. Thus, we decided to use a dummy coding procedure to control for their effects while predicting the variables of the main interest (Hutcheson & Sofroniou, 1999). Primary degree and rural area were used as reference categories.

We have now reanalysed our data using dummy variables (both Study 1 & Study 2). 

The results could be found in the main file.

---

## [Decision Letter · Decision Letter 2]

26 Jul 2022

The Devil is not as Black as He is Painted? On the Positive Relationship Between Food Industry Conspiracy Beliefs and Conscious Food Choices

PONE-D-22-05042R2

Dear Dr. Marchlewska,

We’re pleased to inform you that your manuscript has been judged scientifically suitable for publication and will be formally accepted for publication once it meets all outstanding technical requirements.

Kind regards,

Hans De Steur

Academic Editor

PLOS ONE

Additional Editor Comments (optional):

Reviewers' comments:

Reviewer's Responses to Questions

**Comments to the Author**

1. If the authors have adequately addressed your comments raised in a previous round of review and you feel that this manuscript is now acceptable for publication, you may indicate that here to bypass the “Comments to the Author” section, enter your conflict of interest statement in the “Confidential to Editor” section, and submit your "Accept" recommendation.

Reviewer #2: All comments have been addressed

2. Is the manuscript technically sound, and do the data support the conclusions?

Reviewer #2: Yes

3. Has the statistical analysis been performed appropriately and rigorously? 

Reviewer #2: Yes

4. Have the authors made all data underlying the findings in their manuscript fully available?

Reviewer #2: Yes

5. Is the manuscript presented in an intelligible fashion and written in standard English?

Reviewer #2: Yes

6. Review Comments to the Author

Reviewer #2: Thank you for addressing my concerns and taking the time to make the changes in the analysis. The final manuscript is an interesting view on conspiracy beliefs in Poland.

7. PLOS authors have the option to publish the peer review history of their article (what does this mean?). If published, this will include your full peer review and any attached files.

Reviewer #2: No

---

## [Editor Report · Acceptance letter]

29 Jul 2022

PONE-D-22-05042R2 

The devil is not as black as he is painted? On the positive relations­­­­hip between food industry conspiracy beliefs and conscious food choices 

Dear Dr. Marchlewska:

I'm pleased to inform you that your manuscript has been deemed suitable for publication in PLOS ONE. Congratulations! Your manuscript is now with our production department. 

Kind regards, 

on behalf of

Dr. Hans De Steur 

Academic Editor

PLOS ONE